# Effect of Environmental Heterogeneity and Trophic Status in Sampling Strategy on Estimation of Small-Scale Regional Biodiversity of Microorganisms

**DOI:** 10.3390/microorganisms10112119

**Published:** 2022-10-26

**Authors:** Changyu Zhu, Gaytha A. Langlois, Yan Zhao

**Affiliations:** 1College of Life Sciences, Capital Normal University, Beijing 100048, China; 2Institute of Evolution & Marine Biodiversity, Ocean University of China, Qingdao 266003, China; 3Department of Science and Technology, Bryant University, Smithfield, RI 02917, USA

**Keywords:** equidistant sampling, environmental heterogeneity, trophic status, high-throughput sequencing, microbial diversity

## Abstract

Microorganisms are diverse and play key roles in lake ecosystems, therefore, a robust estimation of their biodiversity and community structure is crucial for determining their ecological roles in lakes. Conventionally, molecular surveys of microorganisms in lakes are primarily based on equidistant sampling. However, this sampling strategy overlooks the effects of environmental heterogeneity and trophic status in lake ecosystems, which might result in inaccurate biodiversity assessments of microorganisms. Here, we conducted equidistant sampling from 10 sites in two regions with different trophic status within East Lake (Wuhan, China), to verify the reliability of this sampling strategy and assess the influence of environmental heterogeneity and trophic status on this strategy. Rarefaction curves showed that the species richness of microbial communities in the region of the lake with higher eutrophication failed to reach saturation compared with that in lower trophic status. The microbial compositions of samples from the region with higher trophic status differed significantly (*P* < 0.05) from those in the region with lower trophic status. The result of this pattern may be explained by complex adaptations of lake microorganisms in high eutrophication regions with environmental conditions, where community differentiation can be viewed as adaptations to these environmental selection forces. Therefore, when conducting surveys of microbial biodiversity in a heterogeneous environment, investigators should incorporate intensive sampling to assess the variability in microbial distribution in response to a range of factors in the local microenvironment.

## 1. Introduction

Shallow lakes are crucial for the conservation of local and global biodiversity [1], they can be classified as oligotrophic, oligo-mesotrophic, mesotrophic, meso-eutrophic, eutrophic, and hypereutrophic classes, based on trophic status according to previous studies [2,3,4,5]. Lake ecosystems vary considerably in species richness, but they contain more biodiversity than other aquatic ecosystems, such as streams, ditches, and temporary ponds [6]. Microorganisms are diverse and play key roles in these ecosystems although their biodiversity varies among different lakes [7,8]. Consequently, investigations focused on microbial diversity are crucial for our understanding of their ecological function in lakes [9,10,11]. In recent years, high-throughput sequencing (HTS) has been widely used in studies of biodiversity in a wide range of aquatic ecosystems including lakes [12,13]. Water samples used for the extraction of metagenomic DNA for estimating the diversity and abundance of biological populations, are usually collected from the same region of a lake or from different regions based on equidistant sampling [14,15,16,17]. This strategy is widely used to monitor density or the abundance of biological populations based on equidistant sites [14,15,16,17,18]. These sampling sites are always distributed equidistantly along the lakeside, or distributed evenly within the lake. However, these investigations of biodiversity collected samples are based on the equidistant sampling strategy, without considering the environmental heterogeneity and trophic status [5,19,20].

Previous studies of aquatic ecosystems revealed differences of microbial biodiversity, apparently linked to the degree of regional heterogeneity, according to different environmental conditions [13,21,22,23,24,25,26]. Environmental heterogeneity was thought to promote species diversity according to previous studies [27,28,29,30]. The niche differentiation concept, which is a part of species sorting theory, suggests that a more heterogeneous environment could support more species through partitioned niche space [27,29], which implies higher biodiversity. For instance, environmental heterogeneity was an important diversity generating factor in the neotropical palm flora, through the process of speciation in neotropical rain forests within a small scale (0.1–100 m) [28]. A previous study [30] also indicated that the diversity of alpine stream biota was positively correlated with environmental heterogeneity stemming from varying hydrological sources. Furthermore, environmental heterogeneity can be characterized by different local trophic status. Previous studies of freshwater-relating diversity to eutrophication were disputable, finding increasing [31,32] or decreasing diversity [33]. Lefranc et al. (2005) indicated that the least micro-eukaryotic biodiversity was found in a eutrophic lake and moderate micro-eukaryotic biodiversity was found in an oligotrophic lake, while the highest biodiversity was found in an oligo-mesotrophic lake in Massif Central [4]. Consequently, the trophic status was also a principal factor in driving biodiversity in lakes, and the relationship between microbial biodiversity and trophic status still needs to be investigated for us to understand the spatial distribution of microbial biodiversity.

As mentioned above, microorganisms are sensitive to environmental heterogeneity within a small scale, and their community composition varies correspondingly [4]. Consequently, there would be a bias when estimating microbial biodiversity in the same regions with different environmental conditions, e.g., trophic status, if a simple equidistant sampling strategy is employed. However, whether environmental heterogeneity and trophic status influence the microbial niche differentiation (species sorting) and alter microbial biodiversity, and whether the equidistant sampling strategy is reliable for estimating regional microbial diversity in heterogeneous environments, remains unclear. Here, we investigate the microbial diversity in two regions of East Lake (Lake Donghu) (30°33′ N, 114°23′ E), a shallow heterogeneous lake in China, with different trophic status (Figure 1) [5,19,34,35]. East Lake is divided into several parts, such as Sha Lake, Shuiguo Lake, Guozheng Lake, Tangling Lake, and Tuan Lake, by artificial dikes [19]. According to previous studies, Sha Lake and Shuiguo Lake were eutrophic regions while Guozheng Lake and Tangling Lake were meso-eutrophic [5,19,34,35]. Consequently, equidistant sampling was used for collecting water samples from these two regions (Sha Lake and Shuiguo Lake, Guozheng Lake, and Tangling Lake, respectively), based on different trophic status. Briefly, water samples were collected based on same distance between two adjacent samples. Regretfully, the artificial dike of Tuan Lake was closed, resulting in the water samples from Tuan Lake not being obtained. The microbial communities included prokaryotes and eukaryotes, and the high-throughput sequencing of the V4 region of 16S rRNA and 18S rRNA gene for prokaryotes and eukaryotes, respectively, were conducted. We hypothesize that environmental heterogeneity and trophic status will influence microbial biodiversity, and the equidistant sampling strategy will not prove to be reliable for microbial diversity estimates in heterogeneous environments.

## 2. Materials and Methods

### 2.1. Sampling and Environmental Information

Ten sampling sites from Sha Lake, Shuiguo Lake, Guozheng Lake, and Tangling Lake, were chosen for our study based on their trophic status. An equidistant sampling strategy was designed based on 10 sites in two regions (eutrophic and meso-eutrophic) of East Lake, Wuhan, China, on 15 January 2019 (Figure 1). The sites 1 to 3 (Sha Lake and Shuiguo Lake) were located in a region with eutrophic status and sites 4 to 10 (Guozheng Lake and Tangling Lake) were located in a region with meso-eutrophic status. At each site, 5 L of surface water was collected and pre-filtered using 200 μm pore-size meshes. A 0.5 L aliquot of pre-filtered sample was filtered onto a 0.22 μm Durapore membrane (Millipore, MA, USA). In addition, a further 0.5 L aliquot of pre-filtered water from each sampling site was mixed and homogenized in a sterilized plastic tank. After that, a 0.5 L of the mixed water was filtered onto 0.22-μm Durapore membrane as a reference sample. A multi-parameter probe (YSI, Yellow Springs, OH, USA) was used to measure dissolved oxygen in situ. Environmental factors including the concentrations of nitrate nitrogen, nitrite nitrogen, ammonia nitrogen, and orthophosphate phosphorus, were measured at each sampling site, as described previously [36].

### 2.2. DNA Extraction, PCR, and High-Throughput Sequencing

Each membrane was moved into a bead tube. Then, environmental DNA was extracted from the membranes using the PowerWater^®^ DNA Extraction kit (Mo Bio, Carlsbad, CA, USA) according to the manufacturer’s instructions. DNA concentration was measured using a NanoDrop 2000 (Thermo Scientific, Waltham, MA, USA). Both the hypervariable regions of 18S and 16S rDNA were amplified from the same total DNA extracts. 18S rDNA fragments (V4 hypervariable region), were PCR amplified with primers EK-565F (5′-GCA GTT AAA AAG CTC GTA GT-3′) and EK-1134R (5′-TTT AAG TTT CAG CCT TGC G-3′) [37]. The PCR mixtures (20 μL) contained 4 μL of 5 × Fastpfu Buffer, 2 μL of dNTPs (2.5 mmol L^–1^), 0.8 μL of each primer (5 μmol L^–1^), 0.4 μL TransStart Fastpfu DNA Polymerase, 0.2 μL BSA and 10 ng of template DNA. The PCR program for eukaryotic primers began with an initial denaturation at 95 °C for 3 min, followed by 30 cycles of 95 °C for 30 s, 55 °C for 30 s, 72 °C for 45 s; and a final extension at 72 °C for 10 min. A primer set (515F/806R) targeting the V4 region of the 16S rRNA gene was used for PCR amplification as described previously [38]. PCR products were sequenced on an Illumina MiSeq platform by Majorbio (Shanghai, China).

### 2.3. Sequence Analysis

High-throughput data analyses were conducted, as described previously [36]. Specifically, raw sequences were demultiplexed, quality filtered by Trimmomatic [39] and merged by FLASH [40] using the default parameters. Reads with exact barcodes and primers, unambiguous nucleotides, and length <200 or >550 for eukaryotes and length <50 or >350 for prokaryotes were retained. Chimeras were identified and removed using UCHIME [41]. Afterward, singleton OTUs (the number of reads was one among all samples) were discarded before the downstream analyses as potential sequencing errors. Remaining sequences were grouped into operational taxonomic units (OTUs, Chicago, IL, USA) for both eukaryotes and prokaryotes at a 97% similarity cutoff using the UPARSE default algorithms [42]. The taxonomy of each OTU was analyzed by RDP Classifier algorithm (http://rdp.cme.msu.edu/, accessed on 22 August 2022) against the Silva database (SSU132) using confidence threshold of 70%. Finally, all samples were rarefied to the same sequence depth (*n* = 23,703 and 35,038 sequences for eukaryotes and prokaryotes, respectively) by random subsampling to standardize sequencing effort.

### 2.4. Statistical Analysis

Statistical analyses and all graphic visualization were performed using the R v.4.0 [43]. Alpha diversity indices, including richness, Shannon-Wiener index, Chao 1 and Pielou’s evenness, were calculated using a “vegan” package [44]. RDA (redundancy analysis) [45] and the Monte Carlo permutation test [46] were performed to explore the relationship between microbial communities and environmental factors. The PCoA (principal coordinate analysis) [45] was performed to visualize patterns of community structures of prokaryotes and eukaryotes based on both Bray-Curtis [47] and unweighted unifrac dissimilarity [48], and the significant differences between regions were tested by running a permutational multivariate analysis of variance (ADONIS) of both eukaryotes and prokaryotes [49,50,51]. The Wilcoxon test [52] was performed to assess significant difference for richness overlaps of samples from regions and reference sample between eutrophic regions and the meso-eutrophic region for prokaryotes and eukaryotes, and differences for concentration of environmental factors between samples from eutrophic and meso-eutrophic regions. Spearman’s rank correlations were performed to explore relationship between richness overlaps of samples from regions and reference sample and concentration of environmental factors for prokaryotes and eukaryotes, respectively [53]. In addition, threshold indicator taxa analyses [54] were performed for selecting indicator OTUs which significantly correlated with change of environmental factors. Indicator taxa with purity and reliability (≥0.95) were plotted in increasing order with respect to their observed environmental change point. The neutral community model (NCM) was used to determine the potential importance of neutral processes on community assembly of samples from different regions [55]. In this model, the R^2^ represents the overall fit to the neutral model, the Nm indicates the metacommunity size (N) times immigration (m). The confidence is 95%, based on 1000 bootstrap replicates.

## 3. Results

### 3.1. Microbial Diversities in Eutrophic and Meso-Eutrophic Regions

After filtering, a total of 2446 and 362 OTUs were detected from prokaryotic and eukaryotic sequences, respectively. The richness, Shannon-Wiener index, Chao 1 and Pielou’s evenness of prokaryotic communities ranged from 480 to 1227, 3.224 to 4.502, 781.110 to 1830.124, and 0.522 to 0.654, respectively. The richness, Shannon-Wiener index, Chao 1 and Pielou’s evenness of eukaryotic communities ranged from 81 to 178, 2.289 to 3.007, 102 to 209.059, and 0.516 to 0.580, respectively (Table 1). Both principal coordinate analysis (PCoA) and permutational multivariate analysis of variance (ADONIS) results showed significant differences (*P* < 0.05) between the community structures of the eutrophic region (site 1, site 2, and site 3) and the meso-eutrophic region (site 4 to site 10) both for prokaryotes and eukaryotes based on Bray-Curtis dissimilarity (Figure 2A) and unweighted unifrac dissimilarity (Figure 2B). In addition, the distances between two adjacent samples from the eutrophic region seems farther than that of the meso-eutrophic region based on both Bray-Curtis and unweighted unifrac dissimilarity, implying that there was stronger niche differentiation in eutrophic region.

### 3.2. Rarefaction Curves and Relative Abundance of Taxonomic Groups in Samples

Rarefaction curves showed a smooth curve both for the reference sample and for samples collected from meso-eutrophic sites, revealing a good coverage of prokaryotic and eukaryotic species number in each case (Figure 3). However, the rarefaction curves for the eutrophic samples were not saturated for both the prokaryote and the eukaryote species number (Figure 3). In addition, Wilcoxon test results showed that there were significant differences of relative abundance value for several taxonomic groups (relative abundance higher than 0.1% in region were shown) between the eutrophic region and the meso-eutrophic region for prokaryotes and eukaryotes (Figure 4). For prokaryotes, the relative abundance of Acidobacteria, Chloroflexi, Omnitrophica, Parcubacteria, Planctomycetes, Verrucomicrobia, and Woesearchaeota_DHVEG-6 for the eutrophic region were significantly (*P* < 0.05) lower than that of the meso-eutrophic region (Figure 4). The relative abundances of other taxonomic groups including Actinobacteria, Bacteroidetes, Cyanobacteria, Euryarchaeota, Firmicutes, Fusobacteria, and Proteobacteria, have no significant differences (*P* > 0.05) between the eutrophic region and the meso-eutrophic region (Figure 4). For eukaryotes, the relative abundances of Apicomplexa, Cercozoa, Fungi, Perkinsea, and Stramenopiles_X for the eutrophic region were significantly lower than meso-eutrophic counterparts (*P* < 0.05). However, the relative abundances of Ciliophora and Ochrophyta for the eutrophic region were significantly higher than that of the meso-eutrophic region. In addition, the relative abundances of Chlorophyta, Cryptophyta, Dinophyta, Katablepharidophy, and Metazoa, between the eutrophic and meso-eutrophic regions, have no significant differences (Figure 4).

### 3.3. The Differences of Species Richness Overlap and Factors Related with Species Richness Overlap

The boxplots for the overlaps of species richness between eutrophic samples and the reference sample and the overlaps of species richness between meso-eutrophic samples and the reference sample, for both prokaryotes and eukaryotes, revealed that samples from the eutrophic region have significantly lower overlap (*P* < 0.05) with the reference sample than do samples from the meso-eutrophic region (Figure 5). For prokaryotes, the overlaps of samples from the eutrophic regions were lower than 0.25, whereas for samples from the meso-eutrophic region the overlaps were generally higher than 0.3. For eukaryotes, the overlaps were lower than 0.5 for samples from the eutrophic region and higher than 0.5 for samples from the meso-eutrophic region.

### 3.4. The Relationship between Microbial Community and Environmental Factors

Redundancy analysis (RDA) results indicated that the microbial communities at the 10 sampling sites were divided into two groups according to the trophic status of the region in which the sites were situated (Figure 6A). Sites 1, 2, and 3 were in the eutrophic region based on the high concentrations of NO_3_-N (nitrate nitrogen), NO_2_-N (nitrite nitrogen), AN (ammonia nitrogen), and PO_4_ (orthophosphate phosphorus). Sites 4 to 10 were in the meso-eutrophic region, based on the high DO (dissolved oxygen) concentration. The Wilcoxon test showed that concentration of DO, and NO_2_-N, were significantly different from the two regions, implying that DO and NO_2_-N were key factors driving microbial community structures (Figure 6B). Several OTUs were strongly associated with environmental factors (Table 2). These included prokaryotic OTUs mainly affiliated with Actinobacteria, Bacteroidetes, Parcubacteria, and Proteobacteria, and eukaryotic OTUs mainly affiliated with Cercozoa, Chlorophyta, Ciliophora, Cryptophyta, Dinophyta, Fungi, Metozoa, Ochrophyta, and unclassified Eukaryota (Table 2). Monte Carlo permutation tests (Table 3) revealed that both DO and NO_2_-N were significantly correlated with prokaryotic community structures (correlation coefficient: 0.850 and 0.675, respectively). Eukaryotic community structures were also significantly correlated with DO and NO_2_-N (0.906 and 0.773, respectively).

### 3.5. Spearman’s Rank Correlation and Threshold Indicator Taxa with Changing of Environmental Factors

According to Spearman’s rank correlation, the overlaps were positively correlated with DO (Figure 7A) and negatively correlated with NO_2_-N (Figure 7A) for prokaryotes (r = 0.80 and −0.88, respectively) and eukaryotes (r = 0.83 and −0.83, respectively).

In addition, threshold indicator taxa analysis (TITAN) results showed several threshold indicator OTUs that were significantly correlated with DO and NO_2_-N, whereas no indicator taxa correlated significantly with other environmental factors (Figure 7B). Indicator OTUs were positively correlated with DO concentration but negatively correlated with NO_2_-N concentration (Figure 7B). For prokaryotes, we found that the abundances of five positive responding indicator OTUs which assigned to Actinobacteria, Cyanobacteria, Proteobacteria, and unclassified bacteria, were increased with DO concentration between 9.660 to 10.555 mg/L. Additionally, 18 negative responding indicator OTUs, which were assigned to Actinobacteria, Cyanobacteria, Firmicutes, Planctomycetes, Proteobacteria, Thaumarchaeota, and unclassified Bacteria, were decreased with NO_2_-N gradient between 0.0110 to 0.0135 mg/L. The point to emphasize here is that out1864 (Cyanobacteria) aoutOTU2025 (unclassified Bacteria) had significant positive correlation with a change of DO concentration and significant negative correlation with a change of NO_2_-N concentration. For eukaryotes, the abundance of six positive response indicator OTUs which assigned to Ciliophora, Katablepharidophyta, Ochrophyta, and Perkinsea, were positively correlated with changes of DO concentrations from 9.660 to 10.555 mg/L. Additionally, 16 negative responding indicator OTUs, which were assigned to Cercozoa, Ciliophora, Cryptophyta, Dinophyta, Katablepharidophyta, Ochrophyta, Perkinsea, Stramenopiles_X, and unclassified Eukaryota, were negatively correlated with changes of NO_2_-N concentrations ranging from 0.0110 to 0.0135 mg/L (Figure 7B, Table 4). Note that OTU134 (Katablepharidophyta), OTU235 (Ochrophyta) and OTU137 (Perkinsea) had significant positive correlation with change of DO concentration and significant negative correlation with change of NO_2_-N concentration.

### 3.6. The NCM Explains Different Community Variation between Samples from Regions with Different Trophic Status

The neutral community model (NCM) was used to explore the potential importance of neutral processes on community assembly of samples from regions with different trophic status. The NCM explained 30% of taxon detection frequency of prokaryotes in the eutrophic region and 54% of taxon detection frequency of prokaryotes in the meso-eutrophic region (Figure 8A). Similarly, the NCM explained the lower taxon detection frequency of eukaryotes in the eutrophic region (37%), than that of eukaryotes in the meso-eutrophic region (72%) (Figure 8B). The NCM results indicates that deterministic processes played a more significant role in structuring the microbial community assembly in the eutrophic region, while the stochastic processes played the most important role in structuring microbial community assembly in the meso-eutrophic region.

## 4. Discussion

Environmental factors and microbial biodiversity from the eutrophic regions were different from the meso-eutrophic regions (Figure 2, Figure 3, Figure 4, Figure 5 and Figure 6), indicating that our study area (East Lake) is an environmentally heterogeneous lake, which was consistent with previous studies concentrating on East Lake [5,19,34,35]. In addition, the relative abundances of microorganisms in regions with different trophic status were different. For instance, the relative abundance of Acidobacteria was higher in the meso-eutrophic region than that of the eutrophic region (Figure 4), which is consistent with a previous study indicating that the abundance of Acidobacteria was abundant in moderately trophic lake, the distribution of Acidobacteria in lakes varies with trophic status [56]. However, the relative abundance of Ciliophora and Ochrophyta in the eutrophic region was significantly higher than that of the meso-eutrophic region (Figure 4), which is consistent with previous studies showing that ciliate abundance increased from oligotrophic to eutrophic lakes [57], and Ochrophyta was always a dominant phytoplankton in eutrophic lake [58], indicating that they prefer the eutrophic environment. Within this heterogeneous lake, consequentially, the influences of sampling sufficiency on microbial biodiversity analysis between eutrophic and meso-eutrophic regions of East Lake showed significant differences based on the equidistant sampling strategy (Figure 3, Figure 4, Figure 5 and Figure 6). The rarefaction curves of microbial communities from the eutrophic region did not reach saturation like that of the meso-eutrophic region (Figure 3), indicating that there were many microbial species that had not been detected from the eutrophic region using the same sampling density (Figure 3). Furthermore, the overlaps of microbial richness between samples from the eutrophic region and the reference sample were significantly lower than that of the meso-eutrophic region (Figure 5), which suggests that the microbial diversity in the eutrophic region was underestimated and could not represent the microbial diversity of this lake ecosystem. Consequently, sample collection based solely on the equidistant sampling strategy is not reliable for estimating regional microbial biodiversity in a heterogeneous lake. The most likely explanation is that in a homogeneous environment, the loss of niche opportunities leads to reduced differentiation of microbial communities, while heterogeneity increases niche partitioning and therefore promotes community differentiation [27,28,29,30]. Furthermore, environmental heterogeneity would also provide shelter and refuges from adverse environmental conditions, which in turn should promote species persistence [59,60]. Our study area, East Lake, has been shown to be an environmentally heterogeneous lake [5,19,34]. Those regions (lakes) of East Lake are affected by different outlets, resulting in different environmental conditions. For instance, Sha Lake and Shuiguo Lake are surrounded by a densely populated area with several sewage outlets extending into the bay, and the water quality is very poor compared to other lakes, while Guozheng Lake and Tangling Lake have much better water quality because they are away from sewage outlets [20,34]. Therefore, different environmental conditions can lead to regional environmental heterogeneity, which can provide additional niche spaces for microorganisms. To obtain complete regional microbial biodiversity, it is necessary to increase the number of samples in such heterogeneous areas.

Additionally, our results indicated that the community structures in meso-eutrophic regions are similar to each other, while the microbial community differentiation in eutrophic regions were obviously differentiated (Figure 2). Consequently, the species sorting may have a stronger effect on the microbial community of samples from the eutrophic region, while neutral process has a stronger effect on the microbial community of samples from meso-eutrophic region (Figure 8). Thus, the results indicated that there were stronger community differences among samples from the eutrophic regions as compared to samples from the meso-eutrophic regions, as a result of stronger niche differentiation in the eutrophic region [61]. The niche differentiation can be influenced by environmental factors (trophic status), as previous studies indicated that the level of eutrophication of the studied waterbodies and the strength of local environmental gradients play a key role in structuring differentiation of phytoplankton community compositions from 50 freshwater bodies in urban areas [62]. Other hydrological studies have shown that the level of eutrophication is correlated with the biodiversity, indicating that eutrophication is an important anthropogenic pressure and reduces biodiversity in multiple groups of organisms in shallow lakes and ponds worldwide [14,63,64,65]. Eutrophication due to nutrient loading and reduced dispersal owing to habitat fragmentation are global scale pressures that are increasingly affecting biodiversity patterns at multiple spatial scales [66]. However, previous studies also indicated that moderate eutrophication would promote biodiversity by providing several niche spaces, while low biodiversity was detected in hypereutrophic and oligotrophic environment because of their rare niche space [4,5,19]. Our investigation showed that there were many microbial species that had not been detected from the eutrophic region (Figure 3), indicating that there were relatively diverse niche spaces in this region. The results of this study reveal that the environmental factors of DO and NO_2_-N concentrations were responsible for niche differentiation in the eutrophic region and drive the variability of the microbial biodiversity and community structure (Figure 6, Table 3). This is consistent with previous studies indicating that nitrogen limitation drives the phytoplankton community structure in East Lake [67,68]. Furthermore, previous studies indicated that Cyanobacteria could produce nitrite (NO_2_-N) during bloom degradation [69] and the concentrations of DO and NO_2_-N were significantly correlated with the size of Cyanobacteria blooms [70]. Here, we present that several indicator OTUs of Cyanobacteria had strong positive and negative relationships with DO and NO_2_-N concentrations, respectively (Table 4). The NO_2_-N and DO conditions implied should therefore enable us to determine the sampling intensity in different regions.

## 5. Conclusions

In summary, traditional microbial surveys based solely on equidistant sampling have largely neglected the influence of environmental heterogeneity and trophic status on biodiversity analysis with insufficient sampling density. In fact, environmental heterogeneity would give rise to niche differentiation and increase the speciation rates of the microbial community. Trophic status also correlates with microbial biodiversity by species sorting. Consequently, the conventional sampling strategy should be reconsidered in ecosystems with small-scale environmental heterogeneities and trophic status, and we should reasonably increase the number of sampling sites according to local environmental conditions in future research.

## Figures and Tables

**Figure 1 microorganisms-10-02119-f001:**
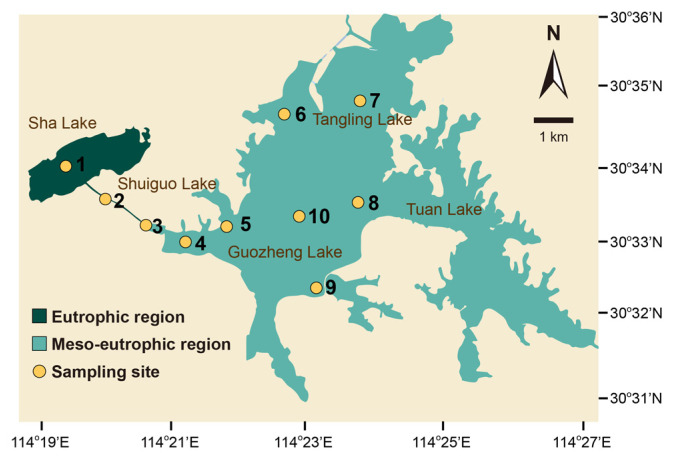
Location of sampling sites. The numbers (1–10) represent the 10 sampling sites.

**Figure 2 microorganisms-10-02119-f002:**
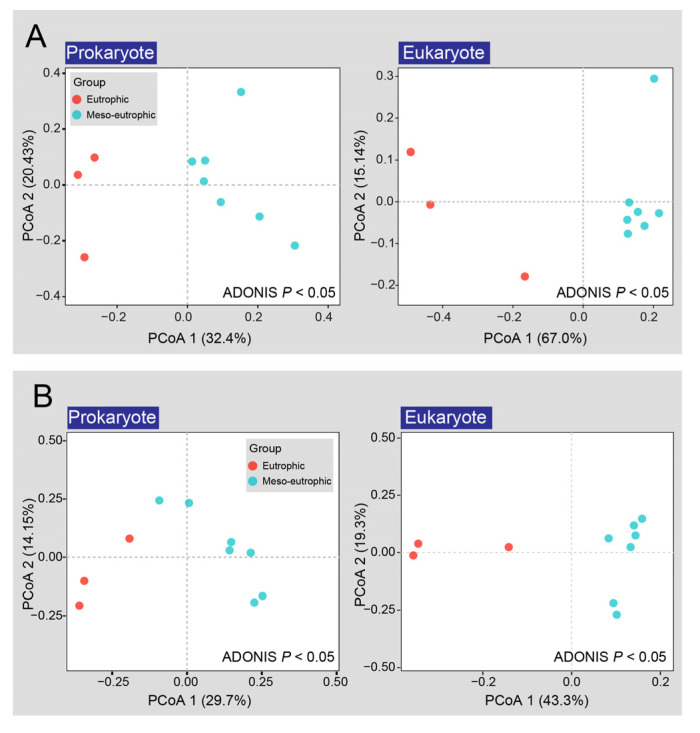
PCoA analysis for samples from eutrophic regions (site 1 to 3) and meso-eutrophic regions (site 4 to 10) based on Bray-Curtis dissimilarity (**A**) and unweighted unifrac dissimilarity (**B**), respectively. *P* represents significance between two regions based on ADONIS analysis for prokaryotes and eukaryotes. *P* < 0.05 is considered as statistically significant.

**Figure 3 microorganisms-10-02119-f003:**
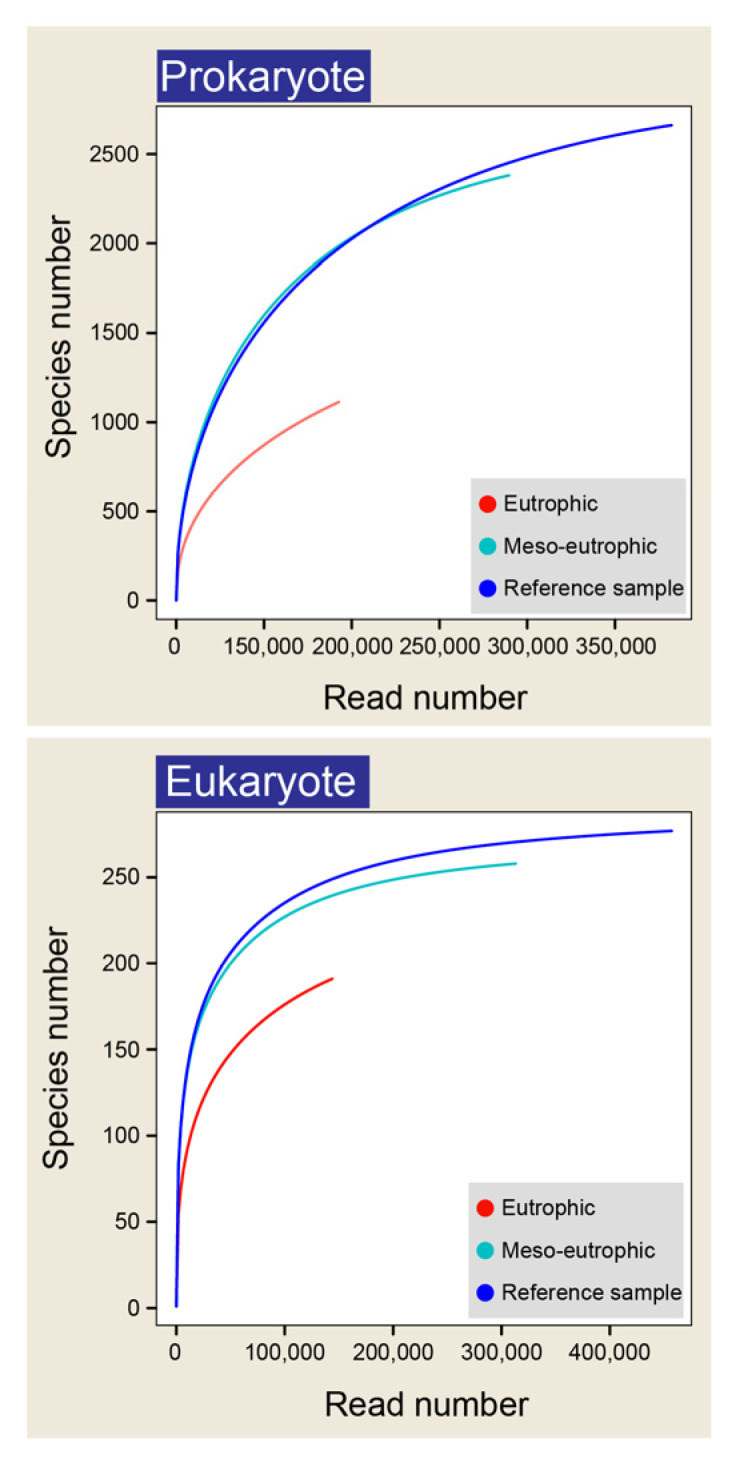
Rarefaction curves of species number in different eutrophic sites based on 97% sequence identity threshold.

**Figure 4 microorganisms-10-02119-f004:**
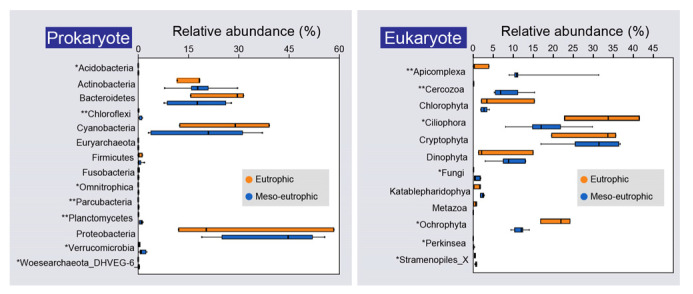
The comparisons for relative abundances of taxonomic groups from eutrophic region and meso-eutrophic region for prokaryotes and eukaryotes, respectively. The value of mean relative abundance higher than 0.1% of taxonomic groups were shown. The significance based on Wilcoxon test. Significant codes: ** *P* < 0.01; * *P* < 0.05.

**Figure 5 microorganisms-10-02119-f005:**
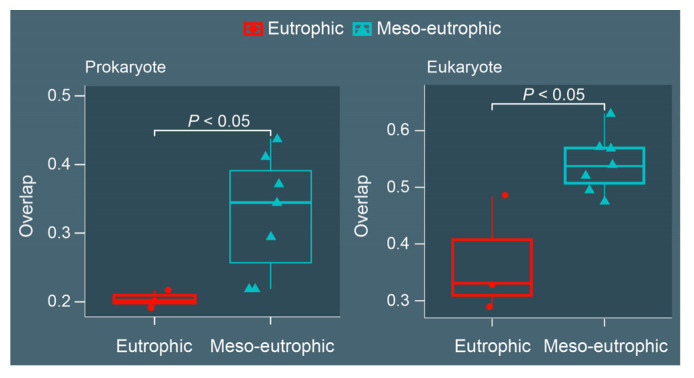
The boxplots for overlaps of richness between eutrophic region and reference data, and between meso-eutrophic region and reference data for prokaryotes and eukaryotes, respectively. The significance based on Wilcoxon test.

**Figure 6 microorganisms-10-02119-f006:**
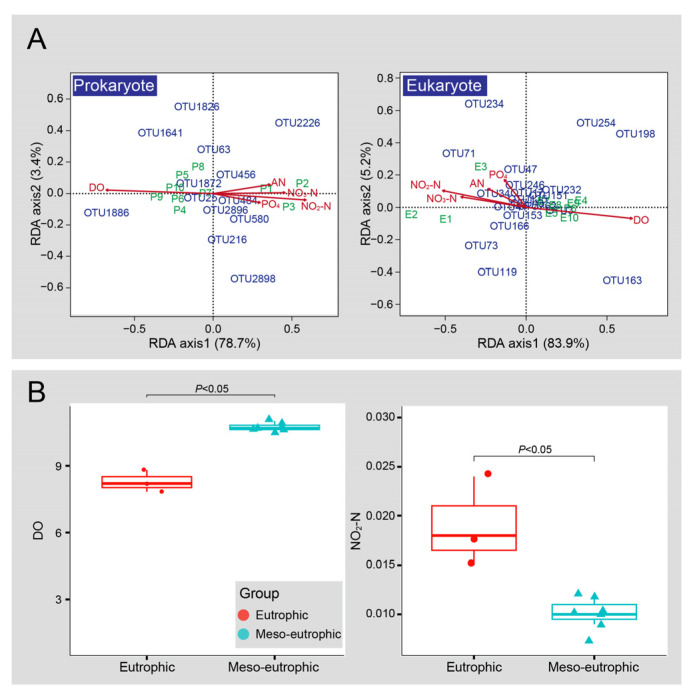
RDA ordination showing the prokaryotic and eukaryotic community structures, respectively (**A**). DO, dissolved oxygen; NO_3_-N, nitrate nitrogen; NO_2_-N, nitrite nitrogen; AN, ammonia nitrogen; PO_4_, orthophosphate phosphorus. The green characters represent 10 samples, P represents prokaryotes from sites and E represents eukaryotes from sites. Red arrows represent measured environmental factors, blue characters represent OTUs that were strongly associated with the first two axes (fitness > 80%) of RDA for prokaryotes and eukaryotes, respectively. Comparison for concentration of DO and NO_2_-N from two regions (**B**). Significance determined by Wilcoxon test. *P* < 0.05 are considered as significant effect.

**Figure 7 microorganisms-10-02119-f007:**
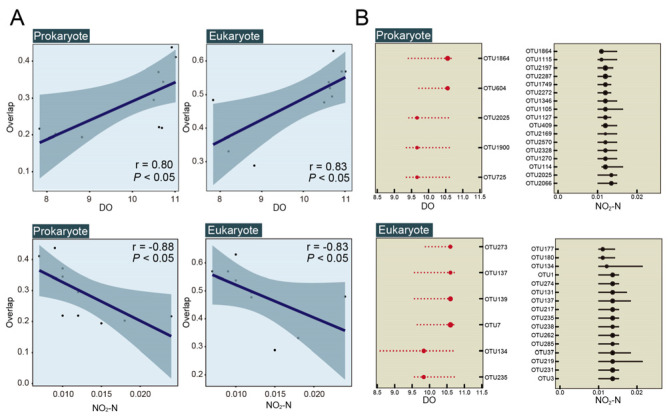
Spearman’s rank correlations between richness overlaps of samples from regions and reference sample and concentration of DO and NO_2_-N for prokaryotes and eukaryotes, respectively (**A**). Threshold indicator taxa analysis (TITAN) of microbial community response to the gradient of DO and nitrite nitrogen (NO_2_-N) (**B**). Black symbols correspond to negative (z−) indicator taxa (i.e., abundance decreases), while red corresponds to positive (z+) indicator taxa (i.e., abundance increases). Symbols are scaled in proportion to z scores. Horizontal lines overlapping each symbol represent 5th and 95th percentiles among 500 bootstrap replicates.

**Figure 8 microorganisms-10-02119-f008:**
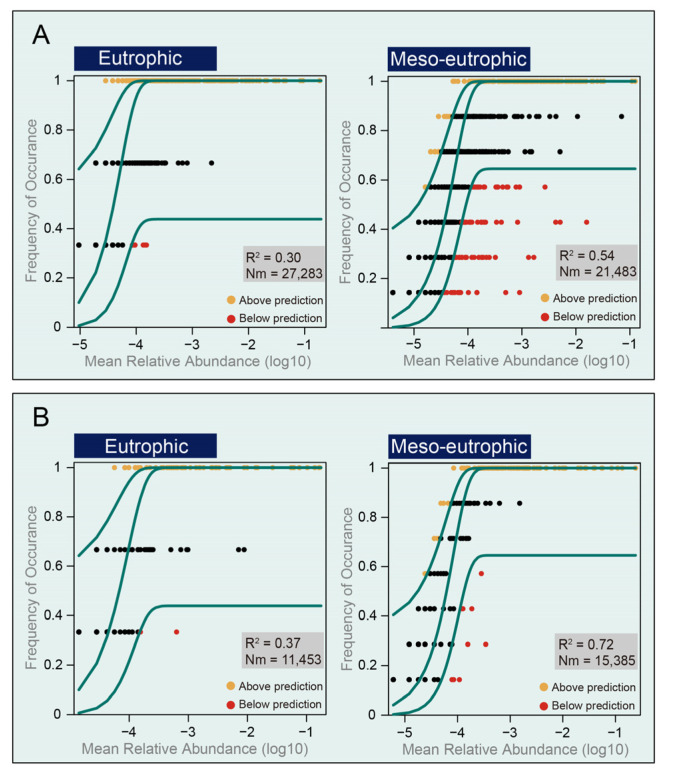
The predicted occurrence frequencies of OTUs from eutrophic and meso-eutrophic regions for prokaryotes (**A**) and eukaryotes (**B**). The solid blue line is the best fit to the neutral community model and the dashed blue line represent 95% confidence intervals around the model prediction. Different colors represent OTUs with more or less frequency than predicted. R^2^ indicates the fit to the neutral model and Nm indicates metacommunity size times immigration.

**Table 1 microorganisms-10-02119-t001:** Diversity of prokaryotic and eukaryotic communities.

Sample	Richness	Shannon-Wiener Index	Chao 1	Pielou’s Evenness	Sample	Richness	Shannon-Wiener Index	Chao 1	Pielou’s Evenness
P1	480	3.224	795.431	0.522	E1	81	2.289	102	0.521
P2	492	3.464	822.694	0.559	E2	95	2.380	127.5	0.523
P3	522	3.694	781.110	0.590	E3	137	2.539	159.8	0.516
P4	537	3.638	931.631	0.579	E4	139	2.737	168.063	0.555
P5	994	4.251	1702.298	0.616	E5	151	2.606	176.143	0.519
P6	771	4.346	1296.606	0.654	E6	137	2.599	178.938	0.528
P7	1130	4.358	1814.052	0.620	E7	161	2.770	198.143	0.545
P8	896	4.374	1488.617	0.643	E8	178	3.007	209.059	0.580
P9	1227	4.502	1830.124	0.633	E9	161	2.849	185	0.561
P10	554	3.874	965.680	0.555	E10	146	2.804	186.4	0.563

P represents prokaryote, E represents eukaryote, 1 to 10 represent site 1 to site 10.

**Table 2 microorganisms-10-02119-t002:** Phylum and genus of each OTU which strongly associated with the first two axes (fitness > 80%) of RDA for prokaryotes and eukaryotes, respectively.

Category	ID	Phylum	Genus
Prokaryote	OTU1886	Actinobacteria	Unclassified Microbacteriaceae
	OTU456	Actinobacteria	Unclassified Sporichthyaceae
	OTU484	Actinobacteria	*Alpinimonas*
	OTU63	Actinobacteria	*Rhodoluna*
	OTU1826	Bacteroidetes	*Sediminibacterium*
	OTU2226	Bacteroidetes	*Flavobacterium*
	OTU2898	Bacteroidetes	*Flavobacterium*
	OTU1872	Parcubacteria	Unclassified Azambacteria
	OTU1641	Proteobacteria	Unclassified Rickettsiales
	OTU216	Proteobacteria	*Methylotenera*
	OTU25	Proteobacteria	Unclassified Comamonadaceae
	OTU2896	Proteobacteria	*Dechloromonas*
	OTU580	Proteobacteria	Unclassified Methylophilaceae
Eukaryote	OTU163	Cercozoa	*Protaspa*
	OTU125	Chlorophyta	*Spermatozopsis*
	OTU198	Ciliophora	*Tintinnidium*
	OTU232	Ciliophora	*Tintinnidium*
	OTU234	Ciliophora	Unclassified Strobilidiidae
	OTU43	Ciliophora	Unclassified Vorticellidae
	OTU71	Ciliophora	*Tintinnopsis*
	OTU119	Cryptophyta	*Teleaulax*
	OTU3	Cryptophyta	*Cryptomonas*
	OTU73	Cryptophyta	*Plagioselmis*
	OTU153	Dinophyta	Unclassified Suessiales
	OTU151	Fungi	Unclassfied Chytridiomycetes
	OTU166	Metazoa	*Parapharyngiella*
	OTU41	Metazoa	*Parapharyngiella*
	OTU17	Ochrophyta	Unclassified Chrysophyceae
	OTU246	Ochrophyta	*Mallomonas*
	OTU254	Ochrophyta	Unclassfied Pedinellales
	OTU34	Ochrophyta	Unclassified Mediophyceae
	OTU47	Ochrophyta	*Chrysosaccus*
	OTU262	Unclassified Eukaryota	

**Table 3 microorganisms-10-02119-t003:** The correlation coefficients between the microbial community structures and environmental factors and result of Monte Carlo permutation test.

Factor	Prokaryote		Eukaryote	
	*r* ^2^	*P*	*r* ^2^	*P*
DO	0.850 *	0.011	0.906 **	0.008
NO_3_-N	0.380	0.175	0.408	0.142
NO_2_-N	0.675 *	0.018	0.773 *	0.013
AN	0.335	0.218	0.449	0.117
PO_4_	0.256	0.438	0.775	0.069

DO, dissolved oxygen; NO_3_-N, nitrate nitrogen; NO_2_-N, nitrite nitrogen; AN, ammonia nitrogen; PO_4_, orthophosphate phosphorus. Significant codes: ** *P* < 0.01; * *P* < 0.05. Permutation: free. Number of permutations: 1000.

**Table 4 microorganisms-10-02119-t004:** Taxonomic assignment of indicator OTU from TITAN for prokaryotes and eukaryotes.

Category	Factor	ID	Phylum	Genus	Env.cp	Purity	Reliability
P	DO	OTU1900	Actinobacteria	*Gordonia*	9.660	0.998	0.972
		OTU1864	Cyanobacteria	Cyanobacteria Subsection III	10.5550	0.996	0.956
		OTU604	Proteobacteria	Unclassified Proteobacteria	10.5550	1	0.992
		OTU725	Proteobacteria	Unclassified Proteobacteria	9.660	1	0.968
		OTU2025	Unclassified Bacteria		9.660	0.998	0.956
	NO_2_-N	OTU1749	Actinobacteria	*Aeromicrobium*	0.0120	1	0.960
		OTU1115	Cyanobacteria	Unclassified Cyanobacteria	0.0110	1	0.956
		OTU1127	Cyanobacteria	Unclassified Cyanobacteria	0.0120	1	0.964
		OTU114	Cyanobacteria	Unclassified Cyanobacteria	0.0120	0.998	0.984
		OTU1864	Cyanobacteria	Cyanobacteria Subsection III	0.0110	1	0.988
		OTU2287	Cyanobacteria	Cyanobacteria Subsection IV	0.0120	1	0.976
		OTU2328	Cyanobacteria	*Baikalospongia*	0.0120	0.996	0.972
		OTU409	Cyanobacteria	Unclassified Cyanobacteria	0.0120	0.998	0.966
		OTU2169	Firmicutes	*Romboutsia*	0.0120	1	0.972
		OTU1270	Planctomycetes	Unclassified Planctomycetaceae	0.0120	0.996	0.956
		OTU2570	Planctomycetes	Unclassified Planctomycetaceae	0.0120	0.998	0.950
		OTU1105	Proteobacteria	Unclassified Proteobacteria	0.0120	1	0.958
		OTU2066	Proteobacteria	Unclassified Alphaproteobacteria	0.0135	1	0.956
		OTU2197	Proteobacteria	Unclassified Proteobacteria	0.0120	1	0.966
		OTU753	Proteobacteria	Unclassified Comamonadaceae	0.0110	0.996	0.952
		OTU1346	Thaumarchaeota	*Nitrosoarchaeum*	0.0120	0.994	0.952
		OTU2025	Unclassified Bacteria		0.0135	0.998	0.954
		OTU2272	Unclassified Bacteria		0.0120	1	0.978
E	DO	OTU139	Ciliophora	Unclassified Prostomatea	9.660	0.994	0.952
		OTU7	Ciliophora	Unclassified Prostomatea	10.5550	1	0.982
		OTU134	Katablepharidophyta	Unclassified Katablepharidales	9.660	0.998	0.970
		OTU231	Ochrophyta	Unclassified Chrysophyceae	9.660	0.990	0.950
		OTU273	Ochrophyta	*Mallomonas*	10.5550	0.996	0.990
		OTU137	Perkinsea	Unclassified Perkinsida	9.660	0.996	0.986
	NO_2_-N	OTU1	Cercozoa	Cercozoa Novel clade 2	0.0135	0.998	0.950
		OTU219	Cercozoa	*Peregrinia*	0.0135	0.996	0.954
		OTU131	Ciliophora	Unclassified Mesodiniidae	0.0135	0.996	0.958
		OTU217	Ciliophora	*Tintinnidium*	0.0135	1	0.986
		OTU274	Ciliophora	Unclassified Prostomatea	0.0135	1	0.962
		OTU177	Cryptophyta	*Cryptomonas*	0.0110	1	0.986
		OTU180	Cryptophyta	*Cryptomonas*	0.0110	0.994	0.956
		OTU3	Cryptophyta	*Cryptomonas*	0.0135	0.998	0.980
		OTU235	Dinophyta	*Prorocentrum*	0.0135	1	0.974
		OTU37	Dinophyta	*Prorocentrum*	0.0135	0.998	0.980
		OTU134	Katablepharidophyta	Unclassified Katablepharidales	0.0120	1	0.982
		OTU231	Ochrophyta	Unclassified Chrysophyceae	0.0135	1	0.978
		OTU238	Ochrophyta	*Pseudopedinella*	0.0135	1	0.960
		OTU137	Perkinsea	Unclassified Perkinsida	0.0135	0.998	0.974
		OTU285	Stramenopiles_X	Unclassified Pseudodendromonadales	0.0135	0.992	0.956
		OTU262	Unclassified Eukaryota		0.0135	1	0.960

P represents prokaryotes; E represents eukaryotes; Env. cp, responding thresholds of OTU; Purity, the proportion of correct assignments as a threshold indicator among bootstrap replicates; Reliability, the reliability proportion of bootstrap replicates at 0.05 significance level.

## Data Availability

The datasets presented in this study can be found in online repositories. The names of the repository/repositories and accession number(s) can be found from the NCBI database: PRJNA765570.

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
