# Peer review of "Effect of Environmental Heterogeneity and Trophic Status in Sampling Strategy on Estimation of Small-Scale Regional Biodiversity of Microorganisms"

_microorganisms, 2022, doi:10.3390/microorganisms10112119_

Round 1

Reviewer 1 Report

Dear the authors, 

Thank you so much for the efforts in conducting this study and writing up the manuscript. I agree that the study raises a very important consideration on studying the microbial community. I would like to first give the brief intake I got from the manuscript. The study was aimed to point out the possible variation of effectiveness of the distance sampling method in identifying microbial community of lake water habitat across different environmental conditions (i.e. level of eutrophication). The results altogether pointed out that distance sampling at the area of low eutrophication was suitable as reflected by microbial community homogeneity in contrast to the area with high eutrophication that showed heterogeneity. Then, if my understanding is correct and the manuscript is aimed to follow this storyline, I have major comments that I would like to ask the authors to consider for the further revisions. 

1. Considering that there are no data from other methods (besides the distance sampling) as the comparison, I think the flow of the storyline might be better if the story was set out from stating that the distance method is applied to two areas of the studied lakes, and then showed the results how the microbial community results from both areas compared to each other. 

2. Following the comment 1, I would advise the authors to add more details of the distance sampling and the background information about the studied lakes in the introduction section. Then, in the methods, elaborate how the sampling was performed following the distance sampling. 

3. I also want to advise the authors to reduce numbers of figures and tables in the manuscript, maybe limit to those that clearly showed differences of the microbial community between the two area. When I read the current version, I get distracted frequently by the context of correlation with environmental parameters (and I think after restructuring the storyline, this issue could be avoided). 

4. In the meantime, I would like to say that the authors have done very good job in the discussion part in pointing out the key messages from the study. However, as I mentioned, restructuring would really help the reader understanding. 

5. Lastly, I would like to also mention about one alternatives. I think all the results can also be reorganized to demonstrate the case study of microbial community investigation in the lake that similar to the one in this study. In overall, that story is going to be really useful for other researchers that they could see what could happen if they study a lake like this with this similar sampling approach. In that case, the authors might need to give extensive introduction on the studied lake and how usually the sampling of lake water for the microbial community analysis is performed.

I hope this is helpful,

Good luck!

Author Response

Response to Reviewer 1 Comments

Microorganisms

October 15, 2022

Re: Effect of environmental heterogeneity and trophic status in sampling strategy on estimation of small-scale regional biodiversity of microorganisms

Thank you for your comments.

We checked and revised our manuscript according to all suggestions. All changes and responses that we’d like to address are as following. Also, some other necessary changes have been done in the new version.

Thank you for your time and effort.

Best regards,

Yan Zhao

Point 1: Considering that there are no data from other methods (besides the distance sampling) as the comparison, I think the flow of the storyline might be better if the story was set out from stating that the distance method is applied to two areas of the studied lakes, and then showed the results how the microbial community results from both areas compared to each other.

Response 1: Thank you for your suggestion, the storyline has been restructured.

Point 2: Following the comment 1, I would advise the authors to add more details of the distance sampling and the background information about the studied lakes in the introduction section. Then, in the methods, elaborate how the sampling was performed following the distance sampling.

Response 2: Thank you for your suggestion, the storyline has been restructured, and the “distance sampling” has been modified to “equidistant sampling”. In addition, the detailed studies of the equidistant sampling and background information about lakes have been added in the new version.

Point 3: I also want to advise the authors to reduce numbers of figures and tables in the manuscript, maybe limit to those that clearly showed differences of the microbial community between the two area. When I read the current version, I get distracted frequently by the context of correlation with environmental parameters (and I think after restructuring the storyline, this issue could be avoided).

Response 3: Thank you for your suggestion, we have rearranged them in the new version.

Point 4: In the meantime, I would like to say that the authors have done very good job in the discussion part in pointing out the key messages from the study. However, as I mentioned, restructuring would really help the reader understanding.

Response 4: Your suggestion really means a lot to us. Yes, it would be more understandable if we restructuring the storyline.

Point 5: Lastly, I would like to also mention about one alternatives. I think all the results can also be reorganized to demonstrate the case study of microbial community investigation in the lake that similar to the one in this study. In overall, that story is going to be really useful for other researchers that they could see what could happen if they study a lake like this with this similar sampling approach. In that case, the authors might need to give extensive introduction on the studied lake and how usually the sampling of lake water for the microbial community analysis is performed.

Response 5: We agree with the reviewer’s assessment. Accordingly, throughout the manuscript, we have revised.

Reviewer 2 Report

The trophic status should be defined at the introduction (eutotrophic or mesotrophic-oligotrophic conditions)

Why the authors chose to amplify the V4 region?

Material and Methods

The authors sampled 10 sites in four lakes… Why the authors did not sample Tuan lake. What was the sampling depth? Surface? I guess the 10 sites will have different depths so it will be interesting to also have a depth profile

It is not clear from here was the reference sample

The sequences should be deposited (e.g. NCBI Sequence Read Archive (SRA))

Alpha diversity and Beta diversity among the two trophic status should be assessed and presented in the results

Results

A global characterization is missing. How many OTU’s were retrieved?   

An exhaustive list of the OTU’s should be presented

The resolution for Eukaryote Is relatively low

Table 3 – I suggest differentiating somehow prokaryotes and eukaryotes in the table

Discusion

The first paragraph reads as results

Maybe explore more the possible functions of the different bacteria found in the different sites – for instance more relative abundance of Bacteroidetes were found at eutrophic sites, why? Same for eukaryotes, more Ciliophora… these patterns should be explored

Author Response

Response to Reviewer 2 Comments

Microorganisms

October 15, 2022

Re: Effect of environmental heterogeneity and trophic status in sampling strategy on estimation of small-scale regional biodiversity of microorganisms

Thank you for your comments.

We checked and revised our manuscript according to all suggestions. All changes and responses that we’d like to address are as following. Also, some other necessary changes have been done in the new version.

Thank you for your time and effort.

Best regards,

Yan Zhao

Point 1: The trophic status should be defined at the introduction (eutotrophic or mesotrophic-oligotrophic conditions)

Response 1: The trophic status in “Materials and methods” has been moved to “introduction”part in the new version.

Point 2: Why the authors chose to amplify the V4 region?

Response 2: For prokaryotes, the primer universality of V4 region is the highest among all high variable regions for 16S rDNA, and the V4 region is still the most widely used and recognized amplicon in large-scale microbial researches. For instance, Earth Microbiome Project (EMP) (Luke et al., 2017; Nature; doi:10.1038/nature24621); For microeukaryotes, V4 and V9 region are mostly used, but database of V4 region for 18S rDNA of some microeukaryotes are more complete than that of V9 region (Margot et al., 2018; Environmental Microbiology; doi:10.1111/1462-2920.13952). Based on reasons above, we chose the V4 region for amplifying.

Material and Methods

Point 3: The authors sampled 10 sites in four lakes… Why the authors did not sample Tuan lake. What was the sampling depth? Surface? I guess the 10 sites will have different depths so it will be interesting to also have a depth profile

Response 3: Several artificial dikes are located among parts of Lake Donghu, we were unable to enter the Tuan lake for sampling when we collected water samples in other lakes because the artificial dike of Tuan lake was closed. As described in the “Materials and methods”, “5 L of surface water was collected and pre-filtered using 200 μm pore-size meshes”. Actually, we also got the depths of 10 sites, their depths are low, ranging from 2.0 to 4.5 m, and there are minor differences among water depth of each site. In addition, our main concern is surface water, so we did not show the information for depth of sites in manuscript.

Point 4: It is not clear from here was the reference sample

Response 4: Reference sample was 0.5 L of the mixed water from ten sampling sites.

Point 5: The sequences should be deposited (e.g. NCBI Sequence Read Archive (SRA))

Response 5: The SRA accession number was located in “Data Availability Statement” section according rule of “Microorganisms”.

Point 6: Alpha diversity and Beta diversity among the two trophic status should be assessed and presented in the results

Response 6: Beta diversity has been characterized by PCoA analysis, Alpha diversity has been added to “Microbial alpha and beta diversities in eutrophic and meso-eutrophic regions” in the new version.

Results

Point 7: A global characterization is missing. How many OTU’s were retrieved?

Response 7: The global characterization was added in “results” in the new version.

Point 8: An exhaustive list of the OTU’s should be presented

Response 8: The information of OTUs was listed in “Table 1” in the new version.

Point 9: The resolution for Eukaryote Is relatively low

Response 9: The resolution for Eukaryote is consistent with Prokaryote according to our check.

Point 10: Table 3 – I suggest differentiating somehow prokaryotes and eukaryotes in the table

Response 10: In Table 3 (Table 4 in new version), P represents prokaryote and E represents eukaryote as shown in table note.

Discussion

Point 11: The first paragraph reads as results

Response 11: The first paragraph of discussion has been polished in the new version.

Point 12: Maybe explore more the possible functions of the different bacteria found in the different sites – for instance more relative abundance of Bacteroidetes were found at eutrophic sites, why? Same for eukaryotes, more Ciliophora… these patterns should be explored.

Response 12: The difference for relative abundance of Bacteroidetes between eutrophic region and meso-eutrophic region was not significant, but the relative abundance of Ciliophora and Ochrophyta has significant differences between eutrophic region and meso-eutrophic region. Consequently, we chose the Acidobacteria, Ciliophora and Ochrophyta for discussion in the new version.

Reviewer 3 Report

Review comments:

‘Effect of environmental heterogeneity and trophic status in sampling strategy on estimation of small-scale regional biodiversity of microorganisms’ by Zhu et.al. seems well conducted and the results could lead to some interesting insights associated with biodiversity and community structure of lake ecosystems. 

Each section of the MS is well described and well structured. The study questions/hypothesis is clearly stated and necessary evidence has been provided to answer the questions/prove the hypotheses. I recommend considering the manuscript for possible publication. However, I suggest the authors to address the below minor comments:     

Abstract:

Please rewrite the abstract and clearly state what the results really show. Please also include your suggestions for the readers.

Introduction:

1. I find many irrelevant citations in the manuscript including introduction part. Please replace them with more recent works.     

2. Please give a brief classification of lake ecosystems otherwise the readers will be confused.

Methods:  

1. Please provide the coordinates of the study area.

2. Line: 149, page 4: Analyses?

Results:

1. RDA analysis, fig. 2: axis2 shows very low variation. Please explain the reason.

2. Please consider to rewrite lines 193-198, page: 6.

Conclusion:

Please add implication of your study.

References: 

Please check the references. 

Author Response

Response to Reviewer 3 Comments

Microorganisms

October 15, 2022

Re: Effect of environmental heterogeneity and trophic status in sampling strategy on estimation of small-scale regional biodiversity of microorganisms

Thank you for your comments.

We checked and revised our manuscript according to all suggestions. All changes and responses that we’d like to address are as following. Also, some other necessary changes have been done in the new version.

Thank you for your time and effort.

Best regards,

Yan Zhao

‘Effect of environmental heterogeneity and trophic status in sampling strategy on estimation of small-scale regional biodiversity of microorganisms’ by Zhu et.al. seems well conducted and the results could lead to some interesting insights associated with biodiversity and community structure of lake ecosystems. 

Each section of the MS is well described and well structured. The study questions/hypothesis is clearly stated and necessary evidence has been provided to answer the questions/prove the hypotheses. I recommend considering the manuscript for possible publication. However, I suggest the authors to address the below minor comments:

Abstract:

Point 1: Please rewrite the abstract and clearly state what the results really show. Please also include your suggestions for the readers.

Response 1: As suggested by the reviewer, the abstract has been polished in the new version.

Introduction:

Point 2: I find many irrelevant citations in the manuscript including introduction part. Please replace them with more recent works.

Response 32:  The irrelevant references had been replaced in the new version.

Point 3: Please give a brief classification of lake ecosystems otherwise the readers will be confused.

Response 3: The classification of lake ecosystems has been added in the “Introduction” part in the new version.

Methods:

Point 4: Please provide the coordinates of the study area.

Response 4: The coordinates of the study area from Figure 1 have been added in the new version.

Point 5: Line: 149, page 4: Analyses?

Response 5: We have added the the full name of TITAN, which is “Threshold Indicator Taxa Analyses” according to previous study (Baker and King, 2010; Methods in Ecology and Evolution; 10.1111/j.2041-210X.2009.00007.x).

Results:

Point 6: RDA analysis, fig. 2: axis2 shows very low variation. Please explain the reason.

Response 6: The axis 1 and axis 2 are the constraint axes (the parts that the explanatory variable can explain). In general, we only select the first 2-3 constraint axes with high (and significant) eigenvalues plotted for the general graph, the first 2 constraint axes always explained the majority of variance, especially the axis 1. In our study, the axis 1 has explained the majority of variance, so the variance which axis 2 explained was low.

Point 7: Please consider to rewrite lines 193-198, page: 6.

Response 7: As suggested by the reviewer, we have rewritten these lines.

Conclusion:

Point 8: Please add implication of your study.

Response 8: The implication of this study is “the conventional sampling strategy should be reconsidered in ecosystems with small-scale environmental heterogeneities and trophic status, and we should reasonably increase the number of sampling sites according to local environmental conditions in future research”.

References:

Point 9: Please check the references.

Response 9: The references have been checked in new version.

Round 2

Reviewer 1 Report

Thank you  for the revisions. 

Author Response

Thank you very much.

Reviewer 2 Report

Thank you for addressing all my questions

Can you please calculate also the CHAO estimator for the alpha diversity?

Author Response

Thank you for your suggestion, the chao 1 has been added in the new version.